# A Robust Method to Automatically Detect Fin Whale Acoustic Presence in Large and Diverse Passive Acoustic Datasets

Elena Schall [1,*] and Clea Parcerisas [2,3]

1   Alfred Wegener Institute for Polar and Marine Research, 27570 Bremerhaven, Germany
2   Flanders Marine Institute (VLIZ), 8400 Oostende, Belgium
3   WAVES Research Group, Department of Information Technology, Ghent University, 9052 Gent, Belgium
*   Correspondence: elena.schall@awi.de

**Abstract:** The growing availability of long-term and large-scale passive acoustic recordings open the possibility of monitoring the vocal activity of elusive oceanic species, such as fin whales (*Balaenoptera physalus*), in order to acquire knowledge on their distribution, behavior, population structure and abundance. Fin whales produce low-frequency and high-intensity pulses, both as single vocalizations and as song sequences (only males) which can be detected over large distances. Numerous distant fin whales producing these pulses generate a so-called chorus, by spectrally and temporally over-lapping single vocalizations. Both fin whale pulses and fin whale chorus provide a distinct source of information on fin whales present at different distances to the recording location. The manual review of vast amounts of passive acoustic data for the presence of single vocalizations and chorus by human experts is, however, time-consuming, often suffers from low reproducibility and in its entirety, it is practically impossible. In this publication, we present and compare robust algorithms for the automatic detection of fin whale choruses and pulses which yield good performance results (i.e., false positive rates < 3% and true positive rates > 76%) when applied to real-world passive acoustic datasets characterized by vast amounts of data, with only a small proportion of the data containing the target sounds, and diverse soundscapes from the Southern Ocean.

**Keywords:** fin whale; *Balaenoptera physalus*; automatic detection; chorus; 20 Hz pulse; kurtosis

## 1. Introduction

Fin whales (*Balaenoptera physalus*) are listed as a vulnerable species by the IUCN Red List of Threatened Species and their recovery from past whaling activities is considered as slow [1], though recent studies give rise to hope that recovery rates might increase [2,3]. Fin whales are widely distributed in the world's oceans with a rather oceanic distribution [4,5], which makes their study challenging. For areas with restricted access, such as the Southern Ocean, population structures and abundances are unknown and their investigation is prioritized by the International Whaling Commission's Southern Ocean Research Partnership (IWC-SORP, https://iwc.int/sorp, accessed on 1 May 2022).

Large-scale and long-term underwater passive acoustic monitoring (PAM) programs have generated and are continuously generating massive amounts of passive acoustic data. This growing availability of especially low-frequency recordings (due to less storage limitations on autonomous devices when sampling at low frequencies, e.g., below 1 kHz) creates opportunities to study fin whales through the monitoring of their vocal activity [6–10]. Fin whales produce low frequency vocalizations (i.e., <100 Hz) which are similar across populations worldwide [2,11–18]. Male fin whales, in particular, produce single vocalizations and songs in form of pulse sequences at high sound pressure levels which can be detected over many tens of kilometers (e.g., >50 km [2,15,19–21]). In areas of high fin whale density, the songs and single vocalizations of numerous fin whale males overlap spectrally and temporally, generating the so-called fin whale chorus [2,10,22,23].

The manual review of vast amounts of passive acoustic data by human experts (i.e., identifying single vocalizations) is time-consuming and often suffers from low reproducibility e.g., [24] and in its entirety, it is practically impossible. Multiple studies report considerable differences between analysts for the annotation of passive acoustic data, namely the identification of single vocalizations within acoustic recordings, with up to >50% disagreement between analysts [24–27]. The agreement between multiple analysis rounds by the same analyst has also been reported to be considerably low (i.e., intra-analyst agreement <50% in >50% of the cases), highlighting the problem of reproducibility for human-annotated datasets [24]. Particularly, the identification of single vocalizations within a background chorus of overlapping vocalizations of the same type, as it is the case for fin whale pulses [26], is considered to be one of the main difficulties of manual analyses, causing high inter- and intra-analyst variabilities in the number of annotations within a specific time frame [24,25].

With prior knowledge on the acoustic characteristics of a specific vocalization or an acoustic feature (e.g., a vocalization chorus), a mathematical approach can be programmed to provide a reproducible measure for the presence of the sought vocalization or feature. Automatic sound detection algorithms can then be tuned to find specific vocalizations based on their acoustic characteristics within ambient and anthropogenic noise and discern these vocalizations from sounds of other species [27–29]. To date, fin whale pulses have been automatically detected with several techniques, mainly employing energy and/or template detectors [16,30–32]. However, these methods were tested mostly on relatively small datasets, which did not represent the full variability of real-world soundscapes inherent to year-round recordings from multiple recording stations [30,32] or their performance has not been assessed completely (e.g., number of missed vocalizations has not been checked [13,31,32]). If methods were tested on a large real-world dataset, a labor-intensive manual post-processing of detections was required to avoid high numbers of false positive detections [16]. Techniques to detect the fin whale chorus have also been described in the literature, but the performance of these techniques has not been evaluated [13,23,33].

Here, we present and evaluate multiple detection algorithms for fin whale choruses and pulses which can be applied to real-world passive acoustic datasets characterized by vast amounts of data, with only a small proportion of the data (e.g., <1%) containing the target sounds, and diverse noise scenarios from the Southern Ocean.

## 2. Material and Methods

### 2.1. Test Datasets

The developed detection algorithms for fin whale chorus and single 20 Hz pulses were tuned and tested with two datasets: The chorus dataset (CDS) and the pulse dataset (PDS). No noise normalization (e.g., subtraction of temporally averaged power spectral density (PSD) values from files) was applied to any of the data.

*CDS*—The CDS consisted of 651 5 min audio files from four different pelagic recording locations within the Atlantic sector of the Southern Ocean, sampling eight different years, and all months of the year in order to guarantee a realistic representation of soundscapes (details on dataset in [34]). The duration of five minutes was chosen due to the sampling schedule of the recordings (e.g., 5 min/h; see [34]) and because the energy of fin whale choruses and noise sources is relatively stable over this time frame [13,23]. Within the CDS one type of fundamental frequency band, hereafter termed the low frequency chorus at 20 Hz (LFC2), and two types of overtone frequency bands, hereafter termed the high frequency chorus at 80 Hz (HFC8) and the high frequency chorus at 90 Hz (HFC9), were identified by a human analyst. In 285 of the 651 files the presence of the LFC2 was logged, in 134 the presence of HFC8 was logged, and in 68 the presence of HFC9 was logged. Additionally, the CDS contained some files featuring 20 Hz pulses within the chorus band, complete or parts of Antarctic blue whale z-calls, the Antarctic blue whale chorus, and electronic, strumming and ambient noise (e.g., related to ice). To speed up computation processes the entire CDS data was decimated to 500 Hz sampling rate.

*PDS*—As the PDS, the library of annotated circum-Antarctic recordings for Antarctic blue and fin whale vocalizations published in Miller et al. (2021) [26] was used. This library consists of 1880.25 h of annotated audio recordings across seven Antarctic sites and 11 site-years, covering multiple months per year. From this library, all fin whale pulse annotations ('Bp20' and 'BP20Plus', for details see [26]) were considered if their low frequency limit was below 20 Hz and their Nist Quick SNR (see Box A1 for explanation of the application of this method) was at least 15 dB in order to avoid tuning the detector on annotations of potential misclassifications of fin whale pulses (resulting in total 15,247 annotations of fin whale 20 Hz-pulses from the originally 20,694 annotations of this vocalization type).

### 2.2. Detection of Fin Whale Chorus

Within the CDS one type of fundamental frequency band, the LFC2, and two types of overtone frequency bands, the HFC8 and HFC9, were detected. Each of these three choruses were considered present if a respective metric exceeded a certain threshold. Three different metrics were computed in order to test their efficiency in determining chorus presence: signal-to-noise ratio (SNR), PSD slope and PSD area (details explained below). The code to the chorus detection methods can be accessed at the public GitLab repository "OZA sound detectors" (https://gitlab.awi.de/oza-sound-detectors/fin-whale-chorus-and-pulse-detection.git, accessed on 25 November 2022).

*SNR*—The spectral energy contained in the dominant frequency bands of fin whale pulses, as from now termed fin level (FL), was calculated for each audio file (Equation (4)). In this study the dominant frequency bands of fin whale pulses were defined as the frequency limits 17–25 Hz for the LFC2, 84–87 Hz for the HFC8, and 96–100 Hz for the HFC9 (Equation (3), Table 1). Welch's PSD estimates with 2048 Discrete Fourier Transform (DFT) points, a Hamming window, and 50% overlap were calculated for each file and averaged over each chorus band (Equations (1) and (2), Figure 1).

$$S(\omega) = pwelch(x) \tag{1}$$

$$S(\omega_0 : \omega_1) = S(\omega) \ \ where \ \ \omega_0 \leq \omega \leq \omega_1 \tag{2}$$

$$S_{band} = S\left(\omega_{f0} : \omega_{f1}\right) \tag{3}$$

$$FL_{band} = \overline{S_{band}} \tag{4}$$

where $x$ is the signal to analyze, $S(\omega)$ is the spectral energy of each frequency band $\omega$, $\omega_0$ and $\omega_1$ are the frequency limits of an arbitrary frequency band, *band* is the name of the chorus band (LFC2, HFC8, HFC9), and $\omega_{f0}$ and $\omega_{f1}$ are the frequency limits of the corresponding chorus band.

**Table 1.** Frequency limits for PSD operations per band. All frequency values in Hz.

| band | Fin Whale | | Low Adjacent Noise | | High Adjacent Noise | | Low Slope Limit | | High Slope Limit | | Bandwidth Limit | |
|------|---------------|---------------|----------------|----------------|----------------|----------------|----------------|----------------|----------------|----------------|------------------|------------------|
| | $\omega_{f0}$ | $\omega_{f1}$ | $\omega_{ln0}$ | $\omega_{ln1}$ | $\omega_{hn0}$ | $\omega_{sh0}$ | $\omega_{sh1}$ | $\omega_{hn1}$ | $\omega_{sl0}$ | $\omega_{sl1}$ | $\omega_{F20P0}$ | $\omega_{F20P1}$ |
| LFC2 | 17 | 25 | 11 | 15 | 30 | NA | NA | 33 | 15 | 19 | NA | NA |
| HFC8 | 84 | 87 | 81 | 83 | 90 | 88 | 93 | 94 | 82 | 86 | NA | NA |
| HFC9 | 96 | 100 | 90 | 94 | 103 | 99 | 104 | 105 | 93 | 98 | NA | NA |
| F20P | 15 | 26 | 10 | 15 | 30 | NA | NA | 80 | NA | NA | 13 | 35 |

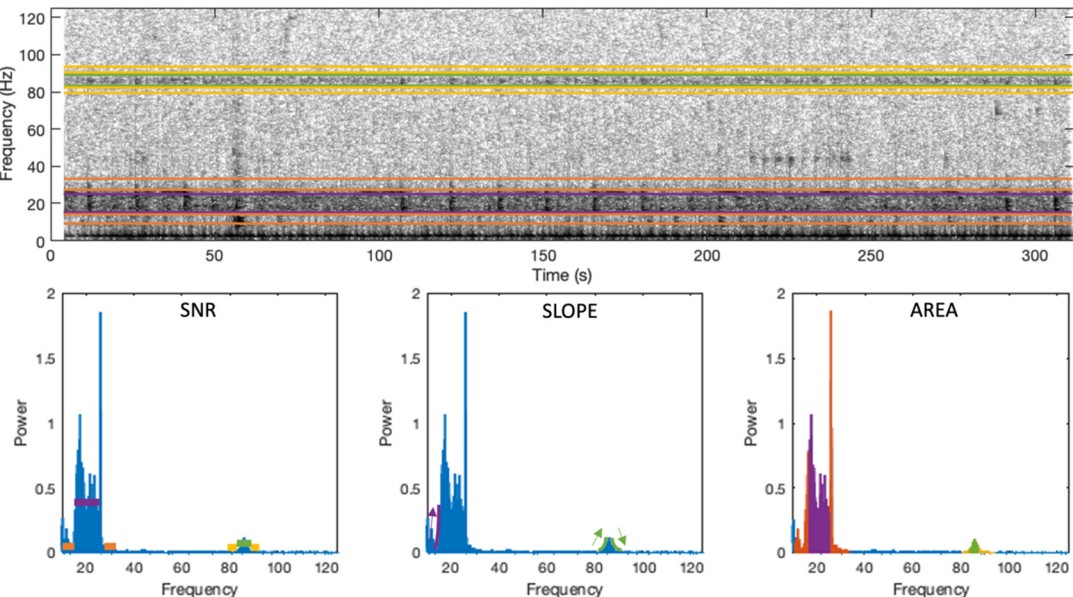

**Figure 1.** Illustration of the temporal-spectral appearance of the fin whale chorus and the three presented methods to detect chorus presence: the SNR method (SNR), the PSD slope method (SLOPE), and the PSD area method (AREA). Purple and green lines/fillings represent the power of the low and high frequency fin whale choruses, respectively, and orange and yellow lines/fillings represent the power of the adjacent respective noise bands.

The noise level (NL) was estimated for the frequency bands 11–15 Hz and 30–33 Hz for the chorus band LFC2, 81–83 Hz and 90–94 Hz for the chorus band HFC8 and 90–94 Hz and 103–105 Hz for the chorus band HFC9 (Equation (5), Table 1). From the above calculated PSD estimates the median of the two respective adjacent frequency bands to each chorus band was computed in order to exclude high energy transient sounds like ice- or animal-borne sounds (Equation (6), Figure 1).

$$S_{band,\ noise} = \begin{bmatrix} S(\omega_{ln0} : \omega_{ln1}) & S(\omega_{hn0} : \omega_{hn1}) \end{bmatrix} \tag{5}$$

$$NL_{band} = median(S_{band,noise}) \tag{6}$$

where $\omega_{ln0}$ and $\omega_{ln1}$ are the frequency limits of the low noise band adjacent to the chorus band and $\omega_{hn0}$ and $\omega_{hn1}$ are the frequency limits of the high noise band adjacent to the chorus band.

The signal to noise ratio for the fin whale chorus energy (SNR$_{band}$; Equation (7)) can then be calculated as:

$$SNR_{band} = 10log_{10}(FL_{band}) - 10log_{10}(NL_{band}) \tag{7}$$

The SNR metric for each of the three chorus bands is calculated for each 5 min audio file and can then be filtered with the determined threshold for the SNR of the chorus (TSNR).

*PSD slope*—The spectral energy of the LFC2 was also estimated as the maximum gradient (i.e., steepness of the rise) of the PSD slope of the audio file between 15 and 19 Hz (Equation (8), Figure 1, Table 1). The descending steepness of the PSD slope at the higher frequency limit of the LFC2 could unfortunately not be used due to the frequent presence of Antarctic blue whale z-calls at the same frequency [35]. The spectral energies of the HFC8 and HFC9 were also estimated as the maximum ascending minus the minimum descending steepness of the PSD slope of the audio file at the respective lower and higher

frequency limits of HFC8 and HFC9 (Equation (8)). These limits were considered 82–86 Hz and 88–93 Hz for HFC8 and 93–98 Hz and 99–104 Hz for HFC9 (see Figure 1, Table 1).

$$SLO_{band} = max(\nabla S(\omega_{sl0} : \omega_{sl1}) - min(\nabla S(\omega_{sh0} : \omega_{sh1})) \tag{8}$$

where $\omega_{sl0}$ and $\omega_{sl1}$ are the lower frequency limits of the chorus for the slope calculation and $\omega_{sh0}$ and $\omega_{sh1}$ are the higher frequency limits of the chorus for the slope calculation. For LCF2 the higher limits where ignored. The PSD slope metric for each of the three chorus bands is calculated for each 5 min audio file and can then be filtered with the determined threshold for the PSD slope of the chorus (TSLO).

*PSD area*—Another parameter computed to describe fin whale choruses was the area under the PSD curve within the frequency band of interest relative to the area occupied by noise (i.e., NL) and chorus (Figure 1). Therefore, the area under the PSD curve of the audio file was integrated between the frequency limits of the chorus (i.e., 17–25 Hz for the LCF2, 84–87 Hz for the HFC8, and 96–100 Hz for the HFC9; Table 1) using the trapezoidal method (Equation (9)). The resulting PSD area was then divided by the area of the PSD of the audio file from the lower frequency limit of the lower adjacent noise band to the upper frequency limit of the upper adjacent noise band (i.e., 11–33 Hz, 81–94 Hz, and 90–105 Hz for LCF2, HFC8, and HFC9, respectively; Table 1), also estimated by the trapezoidal method (Equation (9)).

$$ARE_{band} = \frac{\int S_{band}d\omega}{\int S(\omega_{ln0} : \omega_{hl1})d\omega} \tag{9}$$

The PSD area metric for each of the three chorus bands is calculated for each 5 min audio file and can then be filtered with the determined threshold for the PSD area of the chorus (TARE).

### 2.3. Detection of Fin Whale 20 Hz Pulses

To estimate the presence of fin whales in the vicinity of the recorder (i.e., within a radius of tens of kilometers), it is common to detect fin whale 20 Hz pulses individually (hereafter termed F20P). As their name already indicates, F20P are short (i.e., ~1 s) and loud (i.e., 160–186 dB re 1 μPa at 1 m from the source) impulse sounds between 15 and 28 Hz where the vocalization starts at ~28 Hz and sweeps down to ~15 Hz within approximately 1 s [11,13,15,36–38]. Due to its impulsive nature, it is possible to distinguish between F20P and other sounds, for example Antarctic blue whale Z-calls, by its non-normality of the distribution of its signal samples in contrast to (more) normally distributed signal samples in the case of non-impulse sounds (note that ocean noise is still often non-Gaussian but that only impulse sounds result in comparatively high kurtosis values [39,40]). To test for the normality of a distribution, the shape of the tails of the respective distribution can be calculated as kurtosis (Figure 2) and a threshold value can be applied.

To identify individual F20P in the PDS, audio files were processed in 2s-sliding windows (i.e., with 1.5 s overlap). The resulting 2s-signal was bandpass-filtered (IIR filter order 20) between 15 and 26 Hz (i.e., upper frequency limit at 26 Hz to reduce energy input of blue whale Z-calls [35]) and the kurtosis of the signal was computed (Kurt). The bandpass-filtered signal was also processed with a Teager-Kaiser energy (TKE) operator algorithm to enhance low SNR impulse signals [41]. Kurtosis was then again calculated from the TKE operator modified signal, which together with the original kurtosis value (i.e., as the product of both kurtosis values (KurtProd)) also can be used as an indicator for the presence of a F20P. The timestamp of the maximum value of the by the TKE operator modified signal was used as the center time of the presumed F20P detection. In order to only consider detections, which align with the acoustic characteristics of a F20P, three additional SNR and bandwidth metrics were computed. The SNR of the presumed F20P detection was estimated on the temporal domain by first calculating the root-mean-square of the signal power (i.e., 0.6 s before and after the center time of the presumed F20P) and the root-mean-square of the noise power before and after the signal (i.e., 0.6 s before and

after the start and end of the presumed F20P; Equation (10), Figure 3). The temporal SNR (SNR$_T$) was then calculated as (i.e., formula to calculate temporal SNR when noise does not stop while signal is present [42]):

$$SNR_T = 20log_{10}\left(\frac{\left|rms(x_{signal}) - rms(x_{noise})\right|}{rms(x_{noise})}\right) \tag{10}$$

where $x_{signal}$ is the signal considered in the specified time domain, $x_{noise}$ is the noise considered in the specified time domain (Figure 3) and *rms* stands for the root-mean-squares, computed over the whole bandwidth.

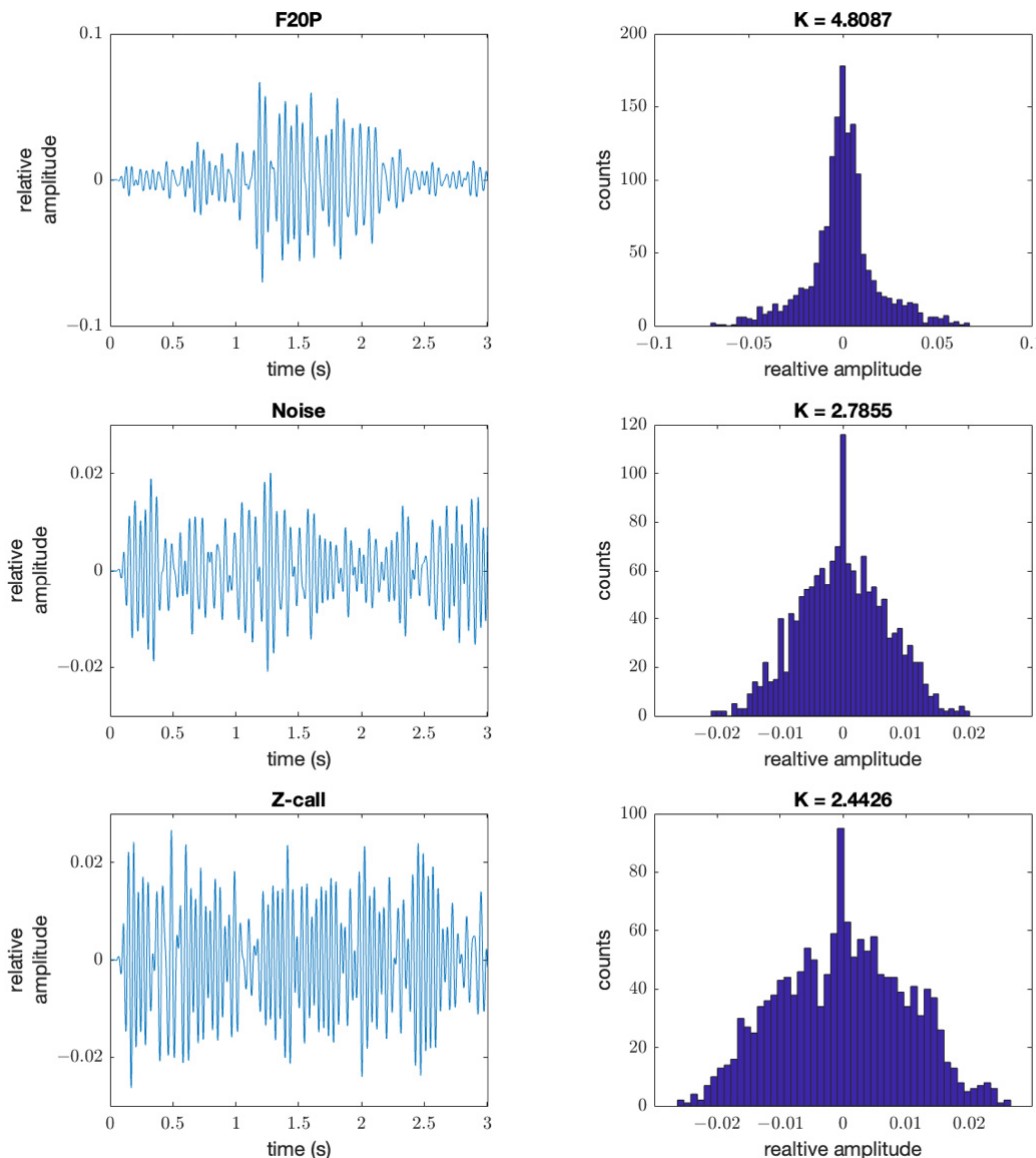

**Figure 2.** Amplitude waveforms and distributions of signals containing a fin whale 20 Hz-pulse (F20P), noise, and an Antarctic blue whale z-call. The respective kurtosis (K) of the amplitude distributions is given above the distribution graph.

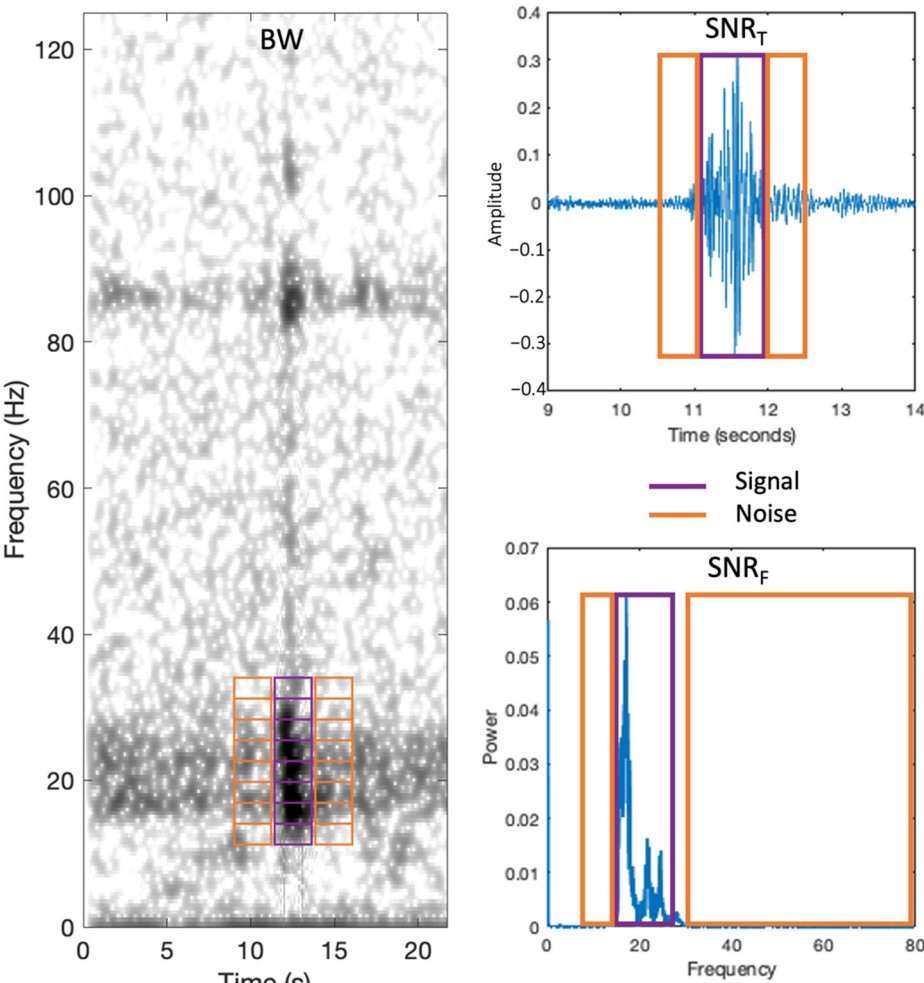

**Figure 3.** Illustration of the temporal-spectral appearance of a fin whale 20 Hz-Pulse and the presented methods to calculate the SNRs and the bandwidth of presumed 20 Hz-Pulses: the bandwidth (BW), the temporal SNR (SNR$_T$), and the spectral SNR (SNR$_F$). Purple boxes represent the power/amplitude of the signal of interest and orange boxes represent the power/amplitude of the adjacent noise.

The SNR of the presumed F20P was also estimated on the frequency domain (SNR$_F$) by calculating the PSD (i.e., with DFT points equal to the signal's sampling rate, a Hamming window, and 50% overlap) of the unfiltered 2s-signal window and averaging the signal's spectral energy between 15 and 26 Hz and the noise's spectral energy between 10 and 15 and 30 and 80 Hz (Equation (11), Figure 3, Table 1):

$$SNR_F = 10log_{10}\left(\overline{S_{F20P}}\right) - 10log_{10}\left(\overline{S_{F20P,noise}}\right) \qquad (11)$$

As a last step, the bandwidth of the detected signal was assessed by calculating the SNR of each frequency bin of the PSD (i.e., with DFT points equal to the signal's sampling rate, a Hamming window, and 50% overlap) of the signal (i.e., 0.6 s before and after the center time of the presumed F20P) compared to the noise before and after the signal (i.e., 0.6 s before and after the start and end of the presumed F20P) in the frequency range between 13 and 35 Hz (Equations (12) and (13), Figure 3, Table 1). The number of frequency bins (total number of frequency bins is 23) with a minimum SNR of 0 dB was then considered as the signal's bandwidth (BW; Equation (14)):

$$S_{signal}(\omega) = pwelch\left(x_{signal}\right) \qquad (12)$$

$$S_{noise}(\omega) = pwelch(x_{noise}) \tag{13}$$

$$BW = sum\left(10log_{10}\left(S_{signal}(\omega_{F20P0} : \omega_{F20P1})\right) - 10log_{10}(S_{noise}(\omega_{F20P0} : \omega_{F20P1})) \geq 0\right) \tag{14}$$

where $S_{signal}$ and $S_{noise}$ are the PSDs of the signal and noise intensities and $\omega_{F20P0}$ and $\omega_{F20P1}$ are the frequency limits of the presumed F20P for the BW calculation.

Due to the fact that all metrics were taken from sliding windows, a single F20P could be detected in multiple windows with varying levels of all metrics. Therefore, all detections within a time period of 2 s were joined by preserving the earliest time stamp and the maxima of kurtosis values, SNRs, and bandwidth. To the resulting timeline of detections, then five different threshold values can be applied to filter detections: the threshold for the signal's kurtosis (TK1), the threshold for the kurtosis product (TK2), the threshold for the temporal SNR (TST), the threshold for the spectral SNR (TSF), and the threshold for the signal bandwidth (TBW, applied in % of the full 23 Hz frequency range). The combination of these thresholds was applied in a Decision-Tree fashion (Figure 4). Furthermore, detections were filtered to preserve only clusters of detections of minimum 5 detections within 5 min, because F20Ps usually occur in sequences [13,15]. To evaluate detector performance, detections with a center time within a 1.5 s range of the start time of an annotated F20P were considered true detections. The code to the F20P detection method can be accessed at the public GitLab repository "OZA sound detectors" (https://gitlab.awi.de/oza-sound-detectors/fin-whale-chorus-and-pulse-detection.git, accessed on 25 November 2022).

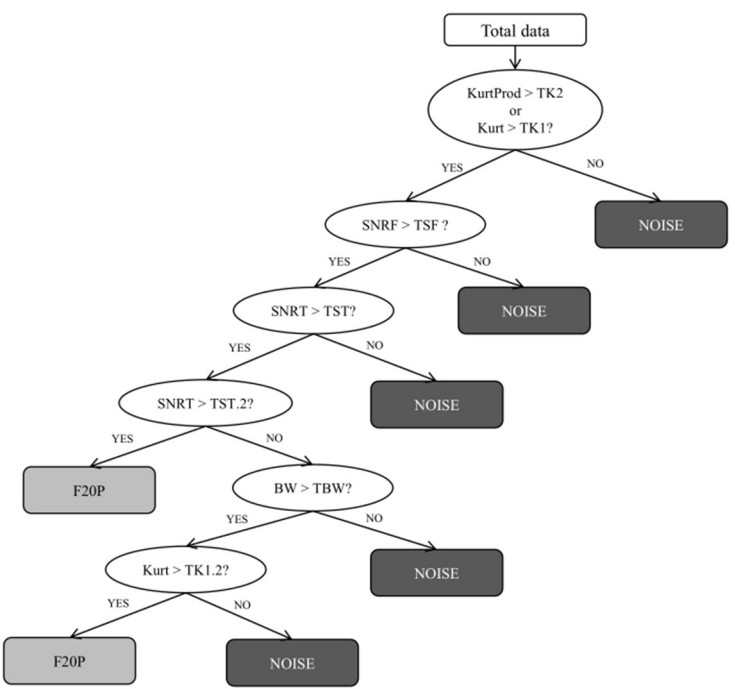

**Figure 4.** Decision tree representation of the applied thresholds (i.e., TK1, TK2, TSF, TST, TST.2, TBW, and TK1.2) to the five metrics of each detection (i.e., KurtProd, Kurt, SNR$_F$, SNR$_T$, and BW).

### 2.4. Threshold Decision

All presented algorithms to detect fin whale chorus and F20P rely on the definition of specific threshold values. The selection of the optimal threshold values was done by maximizing true positive detections while minimizing false positive detections. Assignments of true and false positive detections were based on the comparison of detector output with the annotations from manual screening of the CDS and PDS separately (i.e., chorus presence within 5 min audio files and timestamp annotations of F20P presence, respectively).

In case of the three chorus detection algorithms, three independent thresholds were determined, because the three algorithms represent three distinctly computed metrics for the same characteristic of the chorus band which should be compared to each other. For each algorithm applied to each chorus band a receiver operating characteristic (ROC) curve was plotted and the area-under-the-curve (AUC) value was estimated to evaluate efficiency.

In case of the F20P detection algorithm, a threshold combination for the five computed metrics had to be determined because only the combination of metrics makes it possible to distinguish between true F20P and false F20P detections. The emphasis was put on the reduction in false positives to avoid the over-representation of fin whale pulse presence in real-world PAM datasets by the misclassification of similar sounds. Therefore, all detections were filtered by applying varying combinations of thresholds in multiple steps in a Decision-Tree fashion (Figure 4). First, detections were filtered by allowing for a minimum TK1 or TK2 value, followed by allowing for a minimum TSF value. Then, detections either had to pass a higher TST value (i.e., higher than in the following condition) or detections had to pass a lower TST.2 value (i.e., lower than in the previous condition), but then at the same time pass a minimum TBW value and a higher TK1.2 value (i.e., higher than in the first condition), which allowed for lower SNR detections if these detections covered a certain bandwidth and were clear pulses (ensured by higher kurtosis). True positive and false positive rates were calculated for the entire PDS and for each site-year subset of the PDS by testing in total 423,360 threshold combinations. Additionally, to test the robustness of the method when applied to new data, the method was tested using a blocked cross-validation approach. This means the optimal thresholds were computed using the data of all the sites of the PDS except one, and the performance was evaluated in the site that had not been used to obtain the thresholds.

## 3. Results

The following sections will show the evaluation of detector performance for both chorus and F20P detection in the two real-world passive acoustic datasets from the Southern Ocean (CDS and PDS). Based on this evaluation it is possible to choose appropriate threshold values for the different metrics calculated by the detection algorithms.

### 3.1. Chorus Detection

Fin whale choruses could successfully be detected with all three presented methods, with the PSD slope method being the least efficient compared to the SNR and PSD area methods (Figure 5). The LFC2 was most efficiently detected with the standard SNR method (AUC = 0.98) with which a true positive rate of 89% was achieved when allowing for a 3% false positive rate (Figure 5). The PSD area method yielded similar but a slightly poorer result (AUC = 0.95) with which a true positive rate of 82% was achieved when allowing for a 3% false positive-rate (Figure 5). The PSD slope method performed the poorest with an AUC of only 0.72. The HFC8 was likewise efficiently detected by both the SNR and the PSD area method (AUC = 0.99) with which a true positive rate of 93% was achieved when allowing for a false positive rate of 3% (Figure 5). The PSD slope method only resulted in an AUC of 0.59 for the detection of the HFC8. The HFC9 was only efficiently detected by the PSD area method (AUC = 0.96) with which a true positive rate of 76% was achieved when allowing for a false positive rate of 3% (Figure 5). The SNR and PSD slope method only achieved AUCs of 0.92 and 0.77, respectively.

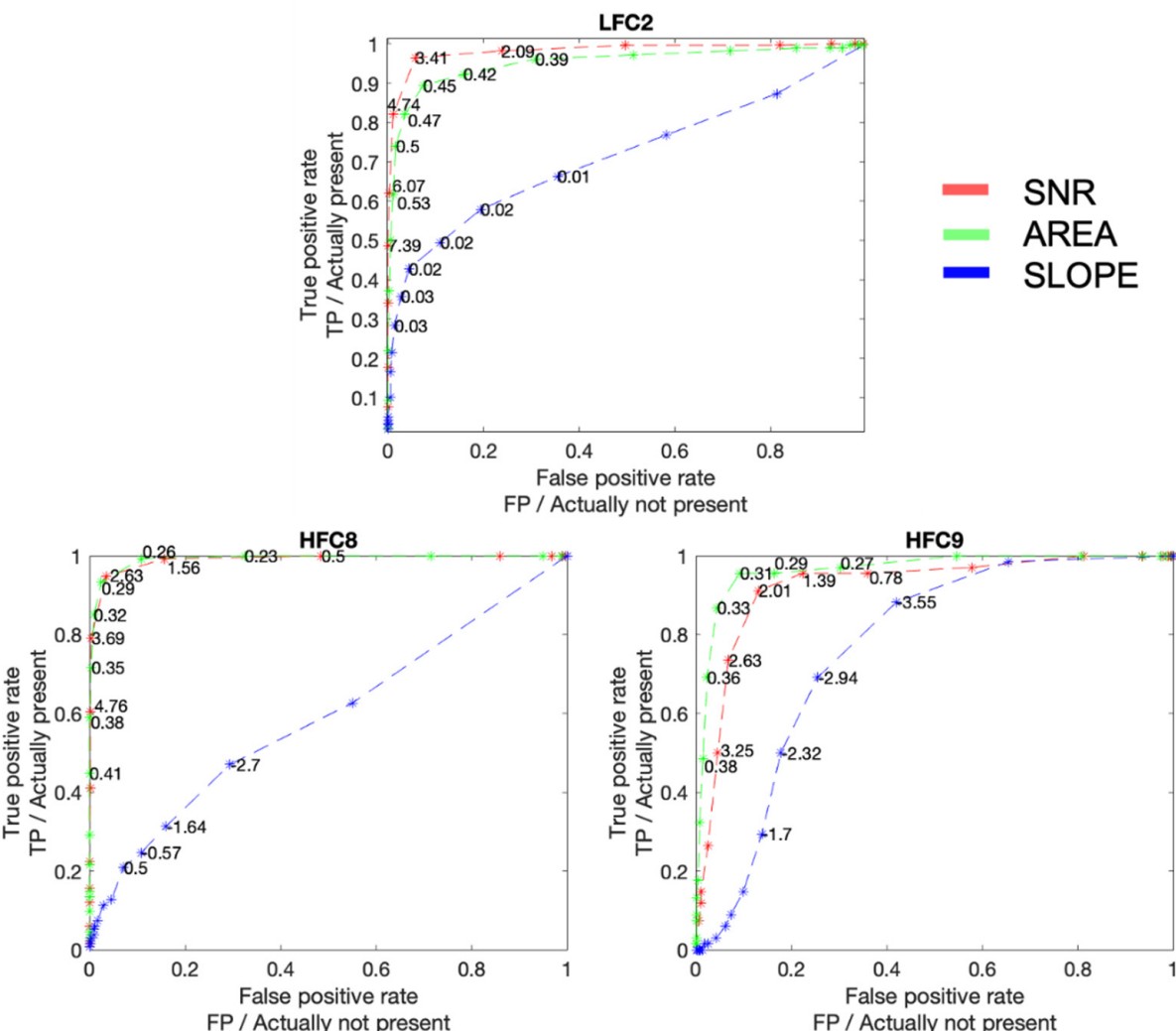

**Figure 5.** Receiver operating characteristics (ROC) curves for the three detection methods, SNR, PSD area, and PSD slope estimating the presence of the three fin whale choruses, LFC2, HFC8, and HFC9. Different threshold values for the respective thresholds, TSNR, TARE, and TSLO are given next to the respective markers in the plots.

*3.2. F20P Detection*

The presented F20P detection algorithm was able to detect these fin whale vocalizations within all site-year subsets of the PDS where F20P presence was annotated by human analysts, while achieving considerably low false positive rates. Allowing for a maximal false positive rate of 1% the F20P detection algorithm achieved a maximum true positive rate of 80% in the entire PDS (Figure 6). The thresholds where the detection algorithm achieved a maximum true positive rate were: TK1 = 3.25, TK2 = 40, TSF = 9, TST = −2, TST.2 = −7, TK1.2 = 4, and TBW = 75).

When different threshold combinations were tested within the framework of a blocked cross-validation, more variable results could be observed (Table 2). We tested with two different threshold combinations, the threshold combination that yielded the 'best' result in each block (i.e., values closest to a true positive rate of 1 and a false positive rate of 0) and the threshold combination that yielded the highest true positive rate at a false positive rate ≤ 1% in each block. At most test locations, true positive and false positive rates were similar to the overall result, except for Elephant Island, where the false positive rates were higher than for the other locations, for Casey and Maud Rise, where the true positive rates were lower than for the entire dataset, and for the Ross Sea, where the false positive rates were considerably lower than for all other locations (Table 2).

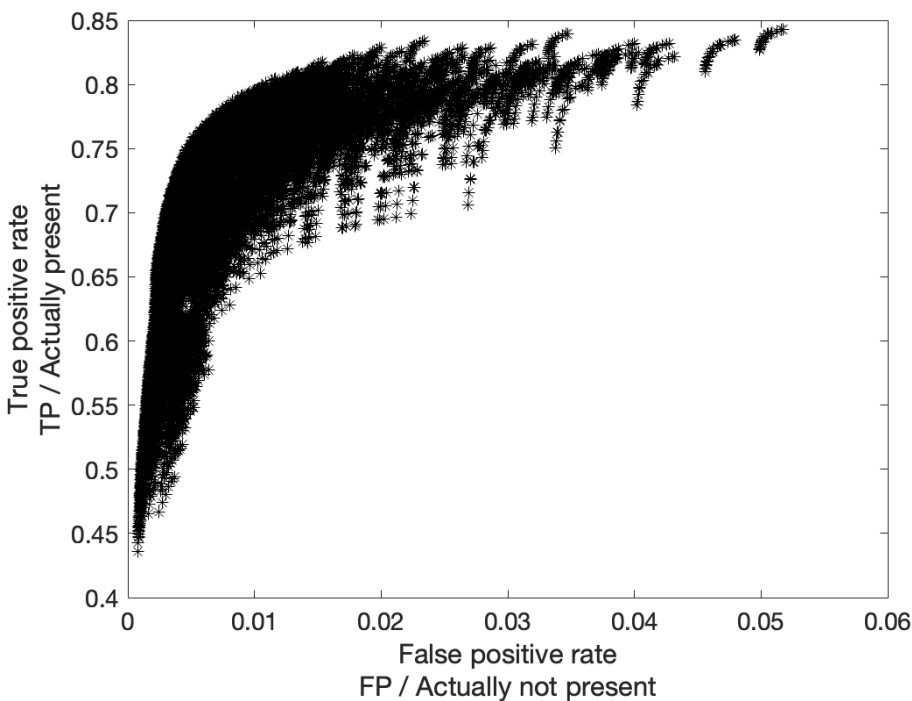

**Figure 6.** Receiver operating characteristics (ROC) curve for the F20P detection algorithm. 423,360 threshold value combinations for the seven different thresholds, TK1, TK2, TSF, TST, TST.2, TK1.2, and TBW were tested and yielded true positive rates up to 80% while false positive rates stayed below 1%.

**Table 2.** Results of the blocked cross-validation between locations for the F20P detection algorithm. Blocks are the combined locations excluding the respective test location. The table provides the number of F20P in the dataset of the test location (N F20P), the threshold combination that yielded the 'best' result in each block (i.e., values closest to a true positive rate of 1 and a false positive rate of 0; Thresholds 'Best'), the resulting true positive rate for the test location (TPR 'Best'), the resulting false positive rate for the test location (FPR 'Best'), the threshold combination that yielded the highest true positive rate at a false positive rate ≤1% in each block (Thresholds '1%'), the resulting true positive rate for the test location (TPR '1%'), and the resulting false positive rate for the test location (FPR '1%').

| Test Location | N F20P | Thresholds 'Best' | | TPR 'Best' | FPR 'Best' | Thresholds '1%' | | TPR '1%' | FPR '1%' |
|---|---|---|---|---|---|---|---|---|---|
| Balleny Island | 321 | TK1 | 2.5 | 0.91 | 0.02 | TK1 | 2.5 | 0.88 | 0.01 |
| | | TK2 | 40 | | | TK2 | 40 | | |
| | | TSF | 6 | | | TSF | 4 | | |
| | | TST | −2 | | | TST | 2 | | |
| | | TST.2 | −8 | | | TST.2 | −8 | | |
| | | TK1.2 | 3.5 | | | TK1.2 | 4 | | |
| | | TBW | 75 | | | TBW | 75 | | |
| Casey | 187 | TK1 | 2.5 | 0.81 | 0.04 | TK1 | 2.5 | 0.75 | 0.008 |
| | | TK2 | 40 | | | TK2 | 40 | | |
| | | TSF | 6 | | | TSF | 8 | | |
| | | TST | −2 | | | TST | −2 | | |
| | | TST.2 | −8 | | | TST.2 | −8 | | |
| | | TK1.2 | 3.5 | | | TK1.2 | 4.25 | | |
| | | TBW | 75 | | | TBW | 80 | | |

**Table 2.** *Cont.*

| Test Location | N F20P | Thresholds 'Best' | | TPR 'Best' | FPR 'Best' | Thresholds '1%' | | TPR '1%' | FPR '1%' |
|---|---|---|---|---|---|---|---|---|---|
| Elephant Island | 10,428 | TK1 | 2.75 | 0.83 | 0.05 | TK1 | 2.5 | 0.79 | 0.027 |
| | | TK2 | 40 | | | TK2 | 40 | | |
| | | TSF | 6 | | | TSF | 9 | | |
| | | TST | −2 | | | TST | 0 | | |
| | | TST.2 | −8 | | | TST.2 | −8 | | |
| | | TK1.2 | 3.5 | | | TK1.2 | 3.75 | | |
| | | TBW | 75 | | | TBW | 75 | | |
| Greenwich | 0 | TK1 | 2.5 | NaN | 0.03 | TK1 | 2.75 | NaN | 0.003 |
| | | TK2 | 40 | | | TK2 | 40 | | |
| | | TSF | 6 | | | TSF | 4 | | |
| | | TST | −2 | | | TST | 2 | | |
| | | TST.2 | −8 | | | TST.2 | −8 | | |
| | | TK1.2 | 3.5 | | | TK1.2 | 4 | | |
| | | TBW | 75 | | | TBW | 75 | | |
| Kerguelen Plateau | 4310 | TK1 | 2.5 | 0.87 | 0.05 | TK1 | 2.5 | 0.82 | 0.009 |
| | | TK2 | 40 | | | TK2 | 40 | | |
| | | TSF | 6 | | | TSF | 9 | | |
| | | TST | −2 | | | TST | −2 | | |
| | | TST.2 | −8 | | | TST.2 | −8 | | |
| | | TK1.2 | 3.5 | | | TK1.2 | 4.25 | | |
| | | TBW | 75 | | | TBW | 75 | | |
| Maud Rise | 1 | TK1 | 2.5 | 0 | 0.05 | TK1 | 2.75 | 0 | 0.011 |
| | | TK2 | 40 | | | TK2 | 40 | | |
| | | TSF | 6 | | | TSF | 4 | | |
| | | TST | −2 | | | TST | −1 | | |
| | | TST.2 | −8 | | | TST.2 | −6 | | |
| | | TK1.2 | 3.5 | | | TK1.2 | 3.75 | | |
| | | TBW | 75 | | | TBW | 80 | | |
| Ross Sea | 0 | TK1 | 2.5 | NaN | 0.001 | TK1 | 2.5 | NaN | 0.00009 |
| | | TK2 | 40 | | | TK2 | 40 | | |
| | | TSF | 6 | | | TSF | 4 | | |
| | | TST | −2 | | | TST | −1 | | |
| | | TST.2 | −8 | | | TST.2 | −8 | | |
| | | TK1.2 | 3.5 | | | TK1.2 | 4 | | |
| | | TBW | 75 | | | TBW | 85 | | |

## 4. Discussion

We presented and evaluated multiple detection methods to detect fin whale chorus and single F20P in large and diverse datasets from the Southern Ocean, which most likely outperform existing methods or proof the viability of previously not evaluated methods.

Detecting both fin whale chorus and F20Ps is advantageous to assess spatio-temporal distribution patterns and (relative) density, because both acoustic signatures provide distinct information on fin whale presence in the study area. Detections of single F20P provide information on concrete vocalization rates per hour/day/month of (individual) whales in relative proximity to the recording device [10,43]. Chorus detection and quantification, in turn, can provide information on approximate vocalization rates of fin whales at larger distances to the recording location [10,13,23,33]. A combination of both a detection method for chorus and single vocalizations in the case of fin whales can therefore help to exploit the full information potential of PAM data.

### 4.1. Chorus Detection

Three different methods for fin whale chorus detection have been tested and evaluated on a large and diverse PAM dataset from the Atlantic sector of the Southern Ocean including four different recording locations, eight different years, and all months of the year. All three

methods are based on the concept of SNR but show variable sensitivities to different acoustic features of the fin whale choruses. The SNR method is a commonly used method in other studies [10,13,23,33] and the evaluation of this method revealed that it is highly efficient for the detection of most choruses. However, for choruses with low SNRs, the PSD area method seems to be more efficient, as it clearly outperformed both other methods in the detection of the HFC9, which was by far the faintest chorus in our test dataset. The minimal SNR of all choruses that could be detected applying the selected thresholds varied between 2.5 and 4 dB re 1 μPa (i.e., LFC2 = 4 dB re 1 μPa, HFC8 = 3 dB re 1 μPa, HFC9 = 2.5 dB re 1 μPa), therefore choruses with SNRs lower than these values would most likely not be detectable with the presented methods. The PSD slope method generally performed rather poor and can therefore not be recommended for the detection of fin whale choruses. Overall, the PSD area method seems to be the most robust method of the three proposed methods as it was the only method which efficiently detected all three types of choruses (i.e., LFC2, HFC8, and HFC9, respectively). In future applications, it could also be explored if a combination of the traditional SNR method with the PSD area method can yield good performance values (i.e., high true positive rate and low false positive rate) for the detection of fin whale choruses in other representative test datasets. The evaluation of performance for automatic detection techniques can only be done by comparing against a ground truth generated by human analysts, which cannot be considered as an absolute truth because the inter- and intra-analyst variability for annotation of PAM data is substantial [24,27]. Due to this, achieved performances of automatic detectors are also limited and reasonable error values have to be allowed (e.g., 3% false positive rate in our case). The acceptance of reasonable error values also has to be considered in any future application of the presented methods, but both the common SNR method and the PSD area method can be used to time-efficiently monitor fin whale acoustic presence in large PAM datasets from all oceans.

When detecting fin whale chorus in PAM datasets from different oceanic regions, frequency limits for the chorus detection might be adapted because these limits can be different among populations and change over time [10,23,30,44,45]. However, the LFC20 seems to be relevant for most fin whale populations and should therefore be a robust indicator for general fin whale presence. The detection of the HFC8, HFC9 and potentially even other HFCs, on the other hand, is most likely population specific [10,44] and can therefore be used to monitor the distribution of acoustic populations and maybe even distinct genetic stocks of fin whales. The detection of HFCs in general can also help to verify that the detection of the LFC2 is not related to the presence of the Antarctic blue whale chorus [13,22]. In the absence of the Antarctic blue whale chorus, in contrast, the frequency limits for the detection of the LFC2 can be adapted to also include slightly higher frequencies (e.g., up to 35 Hz), because the exclusion of the dominant frequency band of Antarctic blue whale Z-calls (i.e., 25–28 Hz [35]) would then not be necessary. In the potential presence of Antarctic blue whale Z-calls, the additional detection of single F20P within the LFC2, as discussed in the following section, would also be another way to confirm the acoustic presence of fin whales.

### *4.2. F20P Detection*

An algorithm for the detection of the most common fin whale vocalization, the F20P, was developed and evaluated which is based on the impulsive nature of this vocalization [15]. Impulsive sounds are characterized by a distribution of their signal samples, which is highly non-Gaussian, in comparison to surrounding rather Gaussian noise. Therefore, kurtosis can be applied as a non-Gaussianity test [39] to distinguish impulsive signals from non-impulsive noise and other non-impulsive vocalizations, such as the Antarctic blue whale Z-call [35]. This method has already been implemented to detect odontocete echolocation clicks [46–48]. However, only using a kurtosis threshold for bandpass-filtered recordings would lead to the detection of any kind of impulsive sound (i.e., non-Gaussian ambient noise such as ice cracking) within the frequency band of fin whale F20Ps. Therefore, both the temporal SNR and the spectral SNR evaluation as well as the assessment of

the signal bandwidth were necessary steps in the implementation of the proposed detection algorithm to avoid large numbers of false positives. Implementing these combined evaluation steps including a two-step threshold check (i.e., allowing also for lower SNR detections when bandwidth and kurtosis are higher, in comparison to higher SNR detections of which additional bandwidth and kurtosis criteria did not need to be met) yielded a robust detection result in a large and very diverse PAM dataset from the Southern Ocean which was specifically compiled to develop and test detection algorithms [26]. A true positive rate of 80% while achieving a false positive rate of <1% is a good result which has not been achieved with other methods when these have been tested on a comparably large and diverse dataset and when manual post-processing by human analysts was not required [13,16,30–32]. The blocked cross-validation showed that the algorithm is robust enough to use with datasets on which thresholds were not tuned explicitly, although some variability in performance results existed among locations and threshold decision also depends on the goal of each specific study (e.g., determination of fin whale acoustic presence or fin whale call rates). Furthermore, the estimated true and false positive rates again have to be evaluated considering the fact that human annotations are not an absolute truth. As discussed earlier, human annotations are prone to a considerable extent of inter- and intra-analyst variability [24,27] which decreases reliability of such ground truth data and is easily identifiable when re-checking annotations for missed vocalizations and potential misclassifications (see Appendix A Figures A1–A6). This reliability of human annotations is then also reflected in the number of false positives and false negatives identified by a detector. The increased number of false positives in the Elephant Island dataset compared to the rest of the PDS, for example, is partly related to missed F20P annotations by the analyst (e.g., Appendix A Figures A1 and A2). The decreased number of true positives in the Casey dataset compared to the rest of the PDS, in turn, is most likely partly related to annotated F20P which are only discernible in the spectrogram due to their energy in the overtone frequency band (i.e., at around 95 Hz, see Appendix A Figure A5). The human error for F20P annotation is most likely especially high, comparable to Antarctic blue whale Z-calls, because these vocalizations easily blend with the mostly concurrently present fin whale chorus which makes their identification challenging [24,25]. The strong chorus band in the Maud Rise dataset, for example, most likely caused the slightly increased number of false positives compared to the rest of the PDS (Appendix A Figure A6). A reduced human error in F20P annotations, for example through independent agreement of multiple analysts on single annotations [24,26], could therefore lead to even better performance results for the proposed algorithm. The proposed detection algorithm based on the principle of kurtosis for the identification of impulsive sounds performs well (i.e., resulting in <1% false positive rate) in the identification of F20P that are at least $-7$ dB above (chorus) background noise and short enough to be impulsive. F20P with particularly long durations (i.e., constant energy over more than 1.5 s) might be a problem to detect with the proposed method as these vocalizations lose their impulsiveness which consequently leads to low kurtosis values. However, F20P with such long durations are most likely very rare, if existent, and have not been documented so far [11,13,15,36–38]. For F20P within the described range of durations (i.e., 0.4–1.2 s) and a considerable SNR (i.e., $-7$ dB), the proposed detection algorithm presents an efficient, robust and reproducible method to estimate fin whale vocalization rates in large and diverse PAM datasets from multiple ocean basins. This is, for example, particularly interesting for acoustic range estimation, the tracking of fin whale locations and the estimation of fin whale density from long-term and large-scale passive acoustic datasets from the Southern Ocean [49,50] or at other remote offshore locations [9,31,51,52].

### 4.3. Comparison with Other Simple Machine Learning Approaches

To be able to apply a method in real datasets, it has to be reliable but also easy to use and implement. Therefore, methods should be simple. The method chosen to detect fin whale chorus was partly already tested in the literature and is a very simple one-step method. Therefore, here this method was only evaluated and no Machine Learning

approaches were tested to determine threshold values. To detect F20P only one metric was not enough, so other approaches were tested before reaching the final model. First, a logistic regression was trained with all the specified F20P metrics (Kurtosis, Kurtosis product, SNR in time domain, SNR in frequency domain and Bandwidth) as independent variables and the detection as the output variable. This obtained a true positive rate of 51.21% when the false positive rate was 1%. A Decision Tree was trained with the same idea, using a Random Search from sklearn [53] to find the best hyperparameters. This solution yield to a true positive rate of 57.03% when the false positive rate was 1%. The final custom-made Decision Tree for threshold values outperformed these methods, while it is also simpler. Therefore, this method was chosen as a better solution that could benefit from the expert's knowledge with the objective of designing a robust system to detect F20P following a similar reasoning to the one a human analyst would use.

### 4.4. Conclusions and Outlook

Fin whale acoustic presence can be monitored by detecting both single vocalizations and fin whale chorus generated by multiple distant individuals vocalizing simultaneously, while the detection of both these sound types generates different knowledge on fin whale distribution, habitat use, migration behavior, and density [9,13,16,23,30–33]. Here, we present techniques to detect both fin whale chorus and single pulses which have been applied to and tested on real-world passive acoustic datasets characterized by vast amounts of data, with only a small proportion of the data containing the target sounds, which represents most available long-term and large-scale PAM datasets. For future application, we plan to analyze our large-scale and long-term dataset from the Atlantic sector of the Southern Ocean [2,22,50,54–56] with the proposed techniques to detect fin whale chorus and single vocalizations. Both detection techniques shall be implemented as feedback loops, which generate probabilities for detections being true positives, because a chorus or a single vocalization present increases the probability of other fin whale sounds being present at the same time, shortly before, or shortly after. This system shall then also be extended with additional detection methods for other fin whale vocalizations which can be used to monitor different types of behavior (i.e., the 40 Hz-Call [57]) and with (existing) methods to detect vocalizations of other species e.g., [22] for the implementation of negative feedback loops in the case of potentially confounding vocalizations of distinct species (e.g., blue and fin whale downsweeps [58]).

The proposed detection techniques can also be implemented separately, in combination or as feedback loops to analyze other PAM datasets from the Southern Ocean and other oceans for the presence of fin whale chorus and F20P. Depending on the spectral characteristics of the respective fin whale vocalizations and noise present, the frequency limits of both algorithms and respective thresholds might need to be adapted in order to guarantee for the desired results. It would be recommended to always test new algorithms which have been developed on a different type of dataset on a small representative subset (e.g., 1% combined from different seasons, years, locations) of the dataset to be analyzed, beforehand, to avoid that unknown sounds in the data cause an unexpectedly high number of false positive detections. Following this approach, the proposed methods present a time-efficient but robust approach for analyzing any PAM data for the acoustic presence of fin whales to aid the development of effective management and conservation strategies for this vulnerable species and associated ecosystems.

**Author Contributions:** Conceptualization, E.S.; methodology, E.S. and C.P.; software, E.S. and C.P.; validation, E.S. and C.P.; formal analysis, E.S.; investigation, E.S.; resources, E.S.; data curation, E.S.; writing—original draft preparation, E.S.; writing—review and editing, E.S. and C.P.; visualization, E.S. and C.P. All authors have read and agreed to the published version of the manuscript.

**Funding:** This research received no external funding.

**Institutional Review Board Statement:** Not applicable.

**Informed Consent Statement:** Not applicable.

**Data Availability Statement:** The code to the two detection methods is published with open access as both MATLAB and Python scripts in the following GitLab repository: https://gitlab.awi.de/oza-sound-detectors/fin-whale-chorus-and-pulse-detection.git, accessed on 25 November 2022.

**Acknowledgments:** Many thanks to the entire team of the ocean acoustics group of the Alfred-Wegener-Institute for the constant support concerning data supply and management, as well as the helpful discussions of methodological approaches. Special thanks to Irene Roca and Victoria Field for providing datasets for initial tests of algorithms. Part of this research was possible thanks to the funding of LifeWatch Belgium.

**Conflicts of Interest:** The authors declare no conflict of interest.

**List of Acronyms**

| | |
|---|---|
| PAM | Passive acoustic monitoring |
| CDS | Chorus dataset |
| PDS | Pulse dataset |
| PSD | Power spectral density |
| LFC2 | Low frequency chorus at 20 Hz |
| HFC8 | High frequency chorus at 80 Hz |
| HFC9 | High frequency chorus at 90 Hz |
| SNR | Signal-to-noise ratio |
| FL | Fin level |
| DFT | Discrete Fourier Transform |
| NL | Noise level |
| $SNR_{band}$ | Signal-to-noise ratio of fin whale chorus energy |
| TSNR | Threshold for the SNR of the chorus |
| TSLO | Threshold for the PSD slope of the chorus |
| TARE | Threshold for the PSD area of the chorus |
| F20P | Fin whale 20 Hz pulses |
| Kurt | Kurtosis of the signal |
| TKE | Teager-Kaiser energy |
| KurtProd | Product of both kurtosis values |
| $SNR_T$ | Temporal SNR |
| $SNR_F$ | SNR on the frequency domain |
| TK1 | Threshold for the signal's kurtosis |
| TK2 | Threshold for the kurtosis product |
| TST | Threshold for the temporal SNR |
| TSF | Threshold for the spectral SNR |
| TBW | Threshold for the signal bandwidth |
| ROC | Receiver operating characteristic |
| AUC | Area-under-the-curve |

**Appendix A**

**Box A1.** Explanation of the NIST Quick method used in this manuscript and available in Raven Pro 1.6. For further details see https://labrosa.ee.columbia.edu/~dpwe/tmp/nist/doc/stnr.txt, accessed on 1 May 2022.

---

**Nist Quick SNR method**
1. Existing temporal-spectral annotation borders were automatically extended to include surrounding noise: 0.6 s were subtracted from the beginning and added to the end of the annotation on the temporal scale. On the spectral scale annotations were consistently extended to cover a bandwidth between 15 and 26 Hz (MATLAB).
2. The RMS power histogram of these annotations were computed in Raven Pro 1.6.
3. The noise level of the annotations was calculated as the 15th percentile of this histogram., the signal level was in contrast calculated as the 85th percentile of the RMS power histogram, and the SNR as the standard ratio between signal and noise.

---

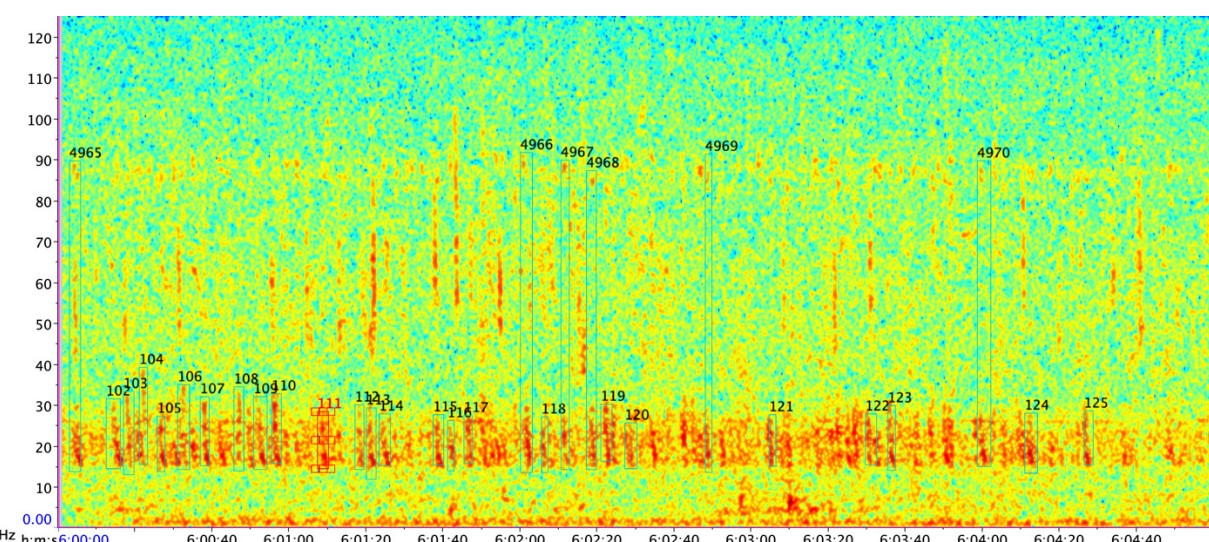

**Figure A1.** F20P log examples from file 20140222_060000_AWI251-01_AU0231_250Hz for the location Elephant Island.

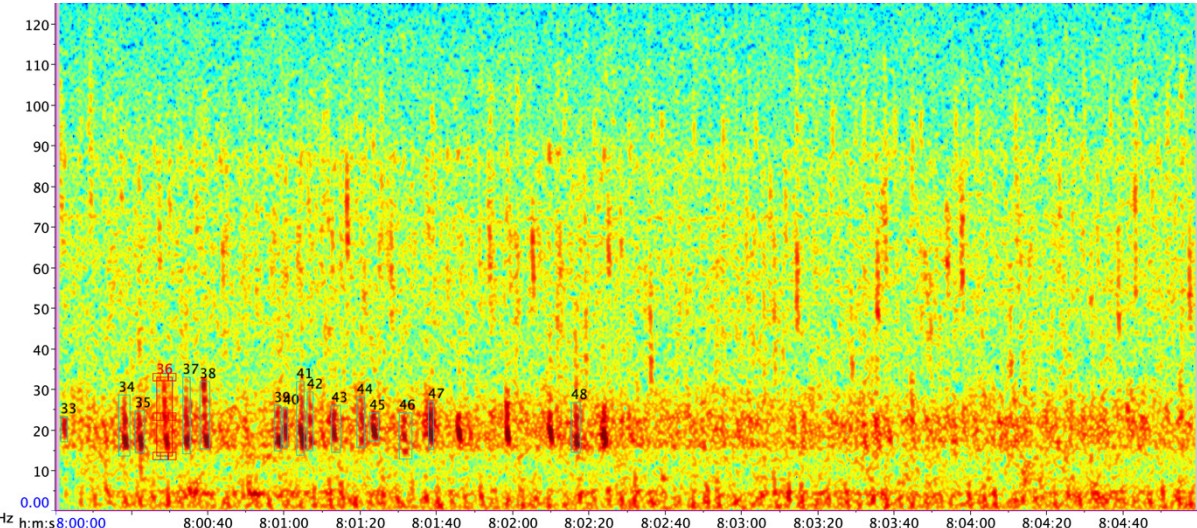

**Figure A2.** F20P log examples from file 20140215_080000_AWI251-01_AU0231_250Hz for the location Elephant Island.

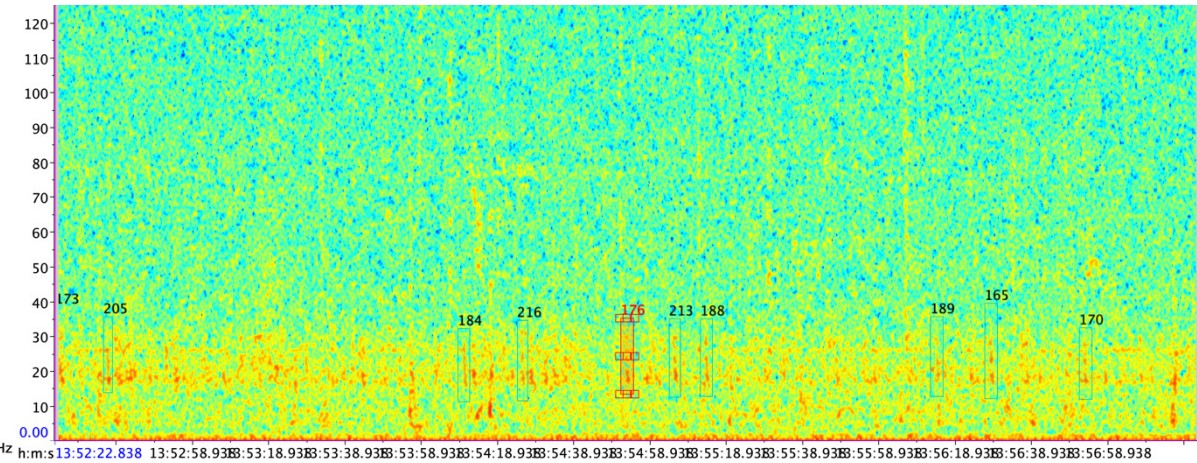

**Figure A3.** F20P log examples from file Balleny20150325_130000 for the location Balleny Island.

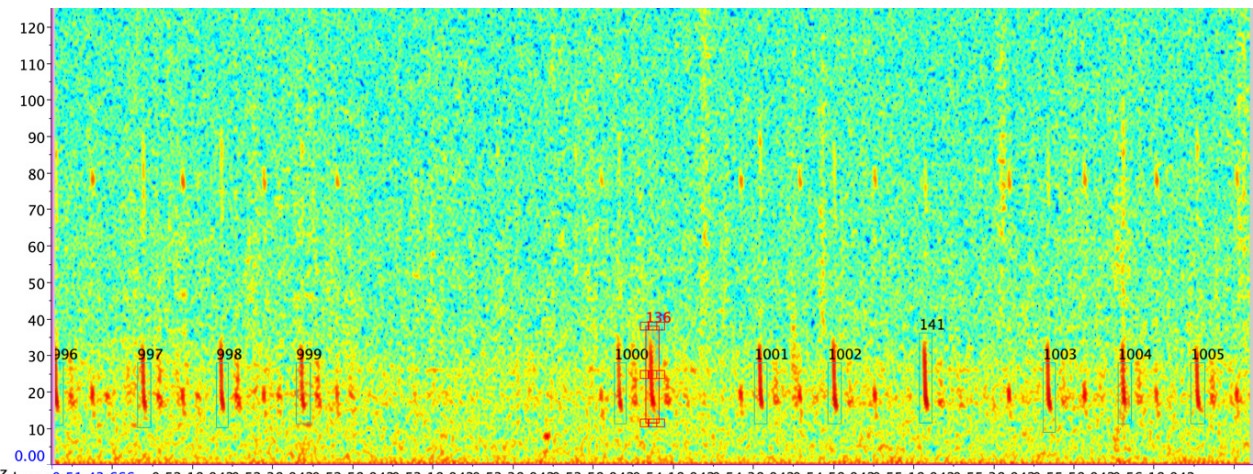

**Figure A4.** F20P log examples from file Balleny20150322_000000 for the location Balleny Island.

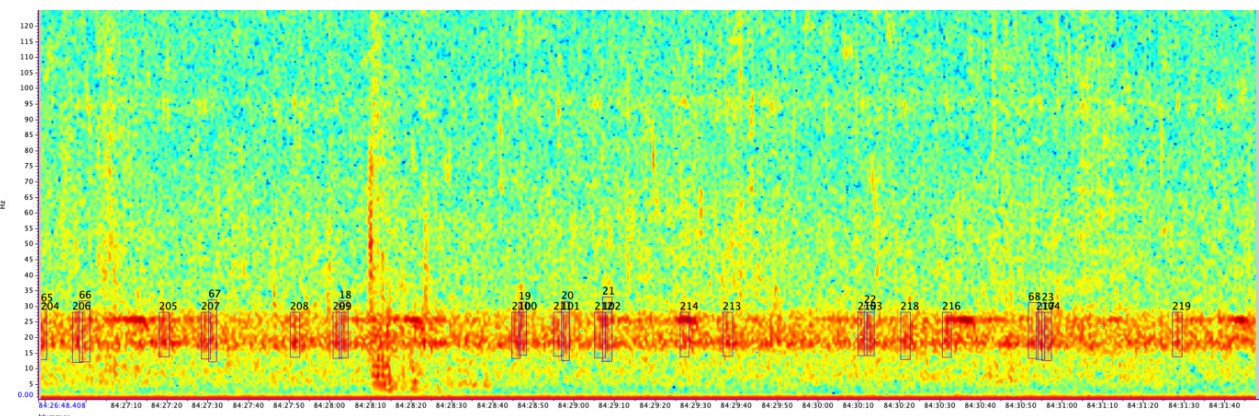

**Figure A5.** F20P log examples from file 20170510_090000 for the location Casey.

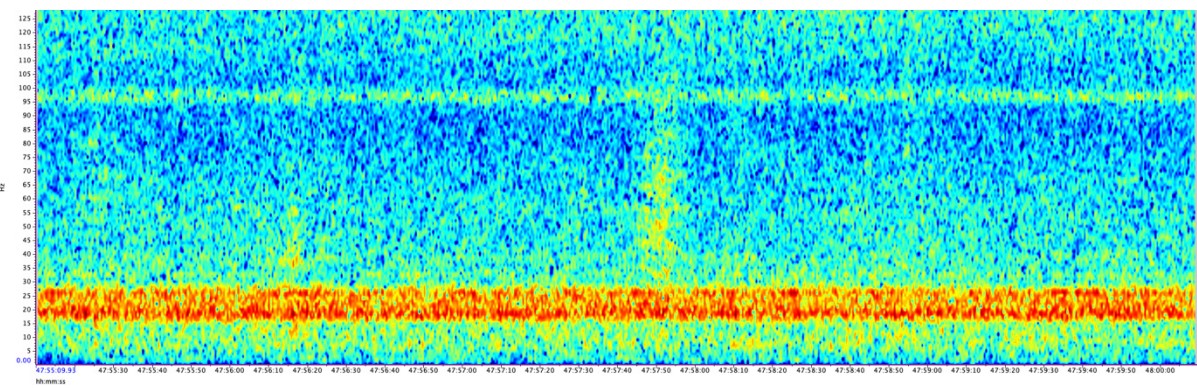

**Figure A6.** F20P log examples from file 20140604_150000 for the location Maud Rise.

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
