# Peer review of "A Robust Method to Automatically Detect Fin Whale Acoustic Presence in Large and Diverse Passive Acoustic Datasets"

_jmse, doi:10.3390/jmse10121831_

Round 1
Reviewer 1 Report
In this paper, the authors present and compare robust algorithms for the automatic detection of fin whale choruses and pulses in large and diverse passive acoustic datasets. In general, the topic in this paper is very interesting. However, the details about authors’ method should be further enhanced before acceptance.
1. In line 137 on page 3, the authors said that ‘Hamming window, and 50% overlap were calculated for each file and averaged over each chorus band’. The authors should clarify why to select 50% overlapping. Besides, some discussions about the overlapping are suggested to be carried out.
2. In practice, the noise distribution plays an important role in detection performance. For ocean noise, some researchers find that the some ocean noise is subjected to non-Gaussian noise such as Middleton noise model [C1] and alpha distribution [C2]. The reviewer wanders to know the detection performance of author’s method in the case of non-Gaussian noise, as this can further enhance the comprehensive study. The authors should discuss this in their discussion part or conclusion part.
[C1]Zhang,et al.Parameter estimation of underwater impulsive noise with the Class B model.IET Radar, Sonar & Navigation,2020,Doi: 10.1049/iet-rsn.2019.0477.
[C2]Mahmood, et al, “Modeling Colored Impulsive Noise by Markov Chains and Alpha-Stable Processes,” in OCEANS 2015 MTS/IEEE, (Genoa, Italy), Doi: 10.1109/OCEANS-Genova.2015.7271550.
3. In line 175, the authors said that ‘The spectral energy of the LFC2 was also estimated as the maximum gradient (i.e., steepness of the rise) of the PSD slope of the audio file between 15 and 19 Hz’. The reviewer wanders to know whether this conclusion, i.e. 15 and 19 Hz, is suitable for all cases.
4. In results part, the authors should further discuss the influence of different SNR on detection performance, as the SNR is the main factor which significantly affects the detection performance.
5. In this paper, some citations are not correct, as the terms ‘Error! Reference source not found’ exit. The authors should proofread the paper before submitting paper.
Reviewer 2 Report
While working through this manuscript I did not discover any serious issues that need to be addressed before this work can be published. I think it is definitely work that should be published and that will be of significant interest to scientists involved in fin whale research.
I suggest all equations should be labelled and that the text refers back to the equations being discussed, making it easier to read the paper. In the version that I read there was an unfortunate error stating "Error! Reference source not found" that appeared all over the place making it a bit difficult to follow along.
The manuscript contains a very long list of acronyms that are unfamiliar to me. I therefore found that I was constantly searching back in the text to find the definitions while reading. With this many acronyms I suggest including an acronym list that the readers can refer back to if needed. Except for the acronyms and the equation labelling, I found the paper well written and with very few typos.
On line 440 the authors claim that their method can be used to time-efficiently monitor fin whale acoustic presence from all oceans. How can they be sure when only one data set was used? How would the choice of frequency bands and thresholds need to be changed in a different location with different additional sound source(s) and general noise? For example, how well would this method work in a place with significant deep-sea vessel noise? Larger vessels are known to radiate lots of sound at the frequencies considered here? Do the authors still think the method would give good results? Perhaps this issue warrants a few sentences?
Round 2
Reviewer 1 Report
The reviewer has no more comments